# Improving Continuous Normalizing Flows
# using a Multi-Resolution Framework

**Vikram Voleti** [1 2]   **Chris Finlay** [3 4]   **Adam Oberman** [3 1]   **Christopher Pal** [5 1 6]

## Abstract

Recent work has shown that Continuous Normalizing Flows (CNFs) can serve as generative models of images with exact likelihood calculation and invertible generation/density estimation. In this work we introduce a Multi-Resolution variant of such models (MRCNF). We introduce a transformation between resolutions that allows for no change in the log likelihood. We show that this approach yields comparable likelihood values for various image datasets, with improved performance at higher resolutions, with fewer parameters, using only 1 GPU.

## 1. Introduction

Reversible generative models derived through the use of the change of variables technique (Dinh et al., 2017; Kingma & Dhariwal, 2018; Ho et al., 2019; Yu et al., 2020) are growing in interest as generative models, because they enable efficient density estimation, efficient sampling, and computation of exact likelihoods. A promising variation of the change-of-variable approach is based on the use of a continuous time variant of normalizing flows (Chen et al., 2018; Grathwohl et al., 2019), which uses an integral over continuous time dynamics to transform a base distribution into the model distribution, called Continuous Normalizing Flows (CNF). CNFs have been shown to be capable of modelling complex distributions such as those associated with images. While this new paradigm for the generative modelling of images is not as mature as Generative Adversarial Networks (GANs) (Goodfellow et al., 2016) or Variational Autoencoders (VAEs) (Kingma & Welling, 2013) in terms of the generated image quality, it is a promising direction of research.

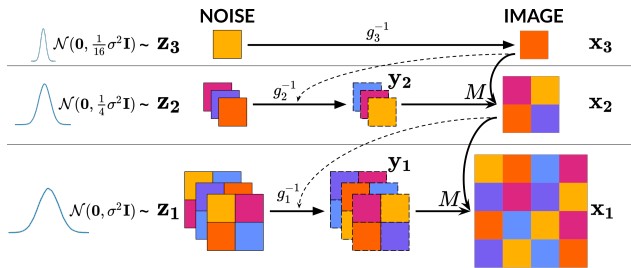

*Figure 1.* The architecture of our MRCNF method (best viewed in color). Continuous normalizing flows (CNFs) $g_s$ are used to generate images $\mathbf{x}_s$ from noise $\mathbf{z}_s$ at each resolution, with those at finer resolutions conditioned (dashed lines) on the coarser image one level above $\mathbf{x}_{s+1}$, except at the coarsest level.

In this work, we focus on making the training of continuous normalizing flows feasible for higher resolution images, and help reduce computation time. We thus introduce a novel multi-resolution technique for continuous normalizing flows, by modelling the conditional distribution of high-level information at each resolution in an autoregressive fashion. We show that this makes the models perform better at higher resolutions. A high-level view of our approach is shown in Figure 1. Our main contributions are:

1. We introduce **Multi-Resolution Continuous Normalizing Flows (MRCF)**, through which we achieve state-of-the-art Bits-per-dimension (BPD) (negative log likelihood per pixel) on ImageNet64 using fewer model parameters relative to comparable methods.

2. We propose a multi-resolution transformation that does not add cost in terms of likelihood.

## 2. Background

### 2.1. Normalizing Flows

Normalizing flows (Tabak & Turner, 2013; Jimenez Rezende & Mohamed, 2015; Dinh et al., 2017; Papamakarios et al., 2019; Kobyzev et al., 2020) are generative models that map a complex data distribution, such as real images, to a known noise distribution. They are trained by maximizing the log likelihood of their input images. Suppose a normalizing flow $g$ produces output $\mathbf{z}$

---
[*]Equal contribution  [1]Mila [2]Université de Montréal, Canada [3]McGill University, Canada [4]Deep Render [5]Polytechnique Montréal, Canada [6]Canada CIFAR AI Chair. Correspondence to: Vikram Voleti <vikram.voleti@gmail.com>.

Third workshop on *Invertible Neural Networks, Normalizing Flows, and Explicit Likelihood Models* (ICML 2021). Copyright 2021 by the author(s).

from an input $\mathbf{x}$ i.e. $\mathbf{z} = g(\mathbf{x})$. The change-of-variables formula provides the likelihood of the image under this transformation as:

$$\log p(\mathbf{x}) = \log \left| \det \frac{\mathrm{d}g}{\mathrm{d}\mathbf{x}} \right| + \log p(\mathbf{z}) \qquad (1)$$

The first term on the right (log determinant of the Jacobian) is often intractable, however, previous works on normalizing flows have found ways to estimate this efficiently. The second term, $\log p(\mathbf{z})$, is computed as the log probability of $\mathbf{z}$ under a known noise distribution, typically the standard Gaussian $\mathcal{N}$.

## 2.2. Continuous Normalizing Flows

Continuous Normalizing Flows (CNF) (Chen et al., 2018; Grathwohl et al., 2019; Finlay et al., 2020) are a variant of normalizing flows that operate in the continuous domain, using the framework of Neural ODEs (Chen et al., 2018). Suppose CNF $g$ transforms its state $\mathbf{v}(t)$ using a Neural ODE with neural network $f$ defining the differential. Here, $\mathbf{v}(t_0) = \mathbf{x}$ is, say, an image, and at the final time step $\mathbf{v}(t_1) = \mathbf{z}$ is a sample from a known noise distribution.

$$\mathbf{v}(t_1) = g(\mathbf{v}(t_0)) = \mathbf{v}(t_0) + \int_{t_0}^{t_1} f(\mathbf{v}(t), t) \, \mathrm{d}t \qquad (2)$$

Chen et al. (2018); Grathwohl et al. (2019) proposed a more efficient method to compute the change in log-probability in the context of CNFs, called the instantaneous variant of the change-of-variables formula:

$$\Delta \log p_{\mathbf{v}(t_0) \to \mathbf{v}(t_1)} = - \int_{t_0}^{t_1} \mathrm{Tr} \left( \frac{\partial f}{\partial \mathbf{v}(t)} \right) \mathrm{d}t \qquad (3)$$

An ODE solver solves both differential equations eq. (2) and eq. (3). Thus, a CNF provides both the final state $\mathbf{v}(t_1)$ as well as the change in log probability $\Delta \log p_{\mathbf{v}(t_0) \to \mathbf{v}(t_1)}$.

Prior works (Grathwohl et al., 2019; Finlay et al., 2020; Ghosh et al., 2020; Onken et al., 2021; Huang & Yeh, 2021) have trained CNFs as reversible generative models of images, by maximizing the likelihood of images:

$$\mathbf{z} = g(\mathbf{x}); \quad \log p(\mathbf{x}) = \Delta \log p_{\mathbf{x} \to \mathbf{z}} + \log p(\mathbf{z}) \qquad (4)$$

where $\mathbf{x}$ is an image, $\mathbf{z}$ and $\Delta \log p_{\mathbf{x} \to \mathbf{z}}$ are computed by the CNF using eq. (2) and eq. (3), and $\log p(\mathbf{z})$ is the likelihood of the computed $\mathbf{z}$ under a known noise distribution, typically the standard Gaussian $\mathcal{N}(\mathbf{0}, \mathbf{I})$. CNF $g$ is trained by maximizing $\mathbb{E}_{\mathbf{x}} \log p(\mathbf{x})$. Novel images are generated by sampling $\mathbf{z}$ from the known noise distribution, and running it through the CNF in reverse.

## 3. Our method

Our method is a reversible generative model of images that builds on top of CNFs. We introduce the notion of multiple resolutions in images, and connect the different resolutions in an autoregressive fashion. This helps generate images faster, with better likelihood values at higher resolutions. Moreover, we used only one GPU in all our experiments. We call this model Multi-Resolution Continuous Normalizing Flow (MRCNF).

### 3.1. Multi-Resolution image representation

Multi-resolution representations of images have been explored in computer vision for decades (Burt, 1981; Witkin, 1987; Burt & Adelson, 1983; Mallat, 1989; Marr, 2010). We express an image $\mathbf{x}$ as a series of high-level information $\mathbf{y}_s$ not present in the immediate coarser images $\mathbf{x}_{s+1}$ (obtained by averaging every 2×2 patch), and a final coarse image $\mathbf{x}_S$:

$$\mathbf{x} \to (\mathbf{y}_1, \mathbf{x}_2) \to \cdots \to (\mathbf{y}_1, \mathbf{y}_2, \ldots, \mathbf{y}_{S-1}, \mathbf{x}_S) \qquad (5)$$

Our overall method is to map these $S$ terms to $S$ noise samples using $S$ CNFs.

### 3.2. Defining the high-level information $\mathbf{y}_s$

The multi-resolution representation in eq. (5) needs to be invertible, i.e. it should be possible to deterministically obtain $\mathbf{x}_s$ from $\mathbf{y}_s$ and $\mathbf{x}_{s+1}$, and vice versa. Further, it is preferable that this transformation incurs minimal additional computational cost, and does not add too much change in log-likelihood. Hence, we choose to perform a linear transformation taking into account the following properties: 1) volume preserving i.e. determinant is 1, 2) angle preserving, and 3) range preserving (respecting the maximum principle, studied for some time, under the notion of *the maximum principle* (Varga, 1966)).

Consider the simplest case of 2 resolutions where $\mathbf{x}_1$ is a 2×2 image with pixel values $x_1, x_2, x_3, x_4$, and $\mathbf{x}_2$ is a 1×1 image with pixel value $\bar{x} = \frac{1}{4}(x_1 + x_2 + x_3 + x_4)$. We require three values $(y_1, y_2, y_3) = \mathbf{y}_1$ that contain information not present in $\mathbf{x}_2$, such that when they are combined with $\mathbf{x}_2$, $\mathbf{x}_1$ is obtained. This could be viewed as a problem of finding a matrix $\boldsymbol{M}$ such that: $[x_1, x_2, x_3, x_4]^\top = \boldsymbol{M}[y_1, y_2, y_3, \bar{x}]^\top$. We fix the last column of $\boldsymbol{M}$ as $[1, 1, 1, 1]^\top$, since every pixel value in $\mathbf{x}_1$ depends on $\bar{x}$. Finding the rest of the parameters can be viewed as requiring four 3D vectors that are (ideally) non-trivially equally spaced. These can be considered as the four corners of a tetrahedron in 3D space, under any rotation in space and scaling of the vectors.

Out of the many possibilities for this tetrahedron, we could choose the matrix that performs the Discrete Haar Wavelet Transform (DHWT) (Mallat, 1989; Mallat & Peyré, 2009). However, this has $\log \left| \det(\boldsymbol{M}^{-1}) \right| = \log(1/2)$, and is therefore not volume preserving. We introduce a variant of the DHWT matrix that is unimodular, i.e. has a determinant of 1 (therefore volume preserving), while also preserving the range of the images for the input and its average:

$$\begin{bmatrix} x_1 \\ x_2 \\ x_3 \\ x_4 \end{bmatrix} = a^{-1} \begin{bmatrix} c & c & c & a \\ c & -c & -c & a \\ -c & c & -c & a \\ -c & -c & c & a \end{bmatrix} \begin{bmatrix} y_1 \\ y_2 \\ y_3 \\ \bar{x} \end{bmatrix} \iff \quad (6)$$

$$\begin{bmatrix} y_1 \\ y_2 \\ y_3 \\ \bar{x} \end{bmatrix} = \begin{bmatrix} c^{-1} & c^{-1} & -c^{-1} & -c^{-1} \\ c^{-1} & -c^{-1} & c^{-1} & -c^{-1} \\ c^{-1} & -c^{-1} & -c^{-1} & c^{-1} \\ a^{-1} & a^{-1} & a^{-1} & a^{-1} \end{bmatrix} \begin{bmatrix} x_1 \\ x_2 \\ x_3 \\ x_4 \end{bmatrix}, \quad (7)$$

where $c = 2^{2/3}$, $a = 4$, and $\log \left| \det(\boldsymbol{M}^{-1}) \right| = \log(1) = 0$. This can be scaled up to larger spatial regions by performing the same calculation for each 2×2 patch. Let $M$ be the function that uses matrix $\boldsymbol{M}$ from above and combines every pixel in $\mathbf{x}_{s+1}$ with 3 corresponding pixels in $\mathbf{y}_s$ to make the 2×2 patch at that location in $\mathbf{x}_s$ using eq. (6):

$$\mathbf{x}_s = M(\mathbf{y}_s, \mathbf{x}_{s+1}) \iff \mathbf{y}_s, \mathbf{x}_{s+1} = M^{-1}(\mathbf{x}_s) \quad (8)$$

### 3.3. Multi-Resolution Continuous Normalizing Flows

Using the multi-resolution image representation in eq. (5), we characterize the conditional distribution over the additional degrees of freedom ($\mathbf{y}_s$) required to generate a higher resolution image ($\mathbf{x}_s$) that is consistent with the average ($\mathbf{x}_{s+1}$) over the equivalent pixel space. At each resolution $s$, we use a CNF to reversibly map between $\mathbf{y}_s$ (or $\mathbf{x}_S$ when $s{=}S$) and a sample $\mathbf{z}_s$ from a known noise distribution. At generation, $\mathbf{y}_s$ only adds information missing in $\mathbf{x}_{s+1}$, but conditional on it. This framework ensures that one coarse image could generate several potential fine images, but these fine images have the same coarse image as their average. This fact is preserved across resolutions.

In principle, any generative model could be used to map between the multi-resolution image and noise. Normalizing flows are good candidates for this as they are probabilistic generative models that perform exact likelihood estimates, and can be run in reverse to generate novel data from the model's distribution. This allows model comparison and measurement of generalization to unseen data. We choose to use the CNF variant of normalizing flows at each resolution, since CNFs have recently been shown to be effective in modeling image distributions using a fraction of the number of parameters typically used in normalizing flows (and non flow-based approaches), and their underlying framework of Neural ODEs have been shown to be more robust than convolutional layers (Yan et al., 2020).

**Training**: We train an MRCNF by maximizing the average log-likelihood of the images in the training dataset under the model, i.e. $\max \mathbb{E}_{\mathbf{x}} \log p(\mathbf{x})$. The log probability of each image $\log p(\mathbf{x})$ can be estimated recursively as:

$$\log p(\mathbf{x}) = \log p(\mathbf{y}_1, \mathbf{x}_2) = \log p(\mathbf{y}_1 \mid \mathbf{x}_2) + \log p(\mathbf{x}_2)$$

$$= \sum_{s=1}^{S-1} \left( \log p(\mathbf{y}_s \mid \mathbf{x}_{s+1}) \right) + \log p(\mathbf{x}_S) \quad (9)$$

where $\log p(\mathbf{x}_S)$ is computed by CNF $g_S$ using eq. (4):

$$\mathbf{z}_S = g_S(\mathbf{x}_S); \quad \log p(\mathbf{x}_S) = \Delta \log p_{\mathbf{x}_S \to \mathbf{z}_S} + \log p(\mathbf{z}_S) \quad (10)$$

and $\log p(\mathbf{y}_s \mid \mathbf{x}_{s+1})$ is also computed by CNFs $g_s$ similarly, conditioning on the coarser image:

$$\begin{cases} \mathbf{z}_s = g_s(\mathbf{y}_s \mid \mathbf{x}_{s+1}) \\ \log p(\mathbf{y}_s \mid \mathbf{x}_{s+1}) = \Delta \log p_{(\mathbf{y}_s \to \mathbf{z}_s)\mid \mathbf{x}_{s+1}} + \log p(\mathbf{z}_s) \end{cases} \quad (11)$$

This model could be seen as a stack of CNFs connected in an autoregressive fashion. Typically, likelihood-based generative models are compared using the metric of bits-per-dimension (BPD), i.e. the negative log likelihood per pixel in the image:

$$\text{BPD}(\mathbf{x}) = \frac{-\log p(\mathbf{x})}{\dim s(\mathbf{x})} \quad (12)$$

Hence, we train our MRCNF to minimize the average BPD of the images in the training dataset, computed using eq. (12). Although the final log likelihood $\log p(\mathbf{x})$ involves sequentially summing over values returned by all $S$ CNFs, each CNF can be trained independently, in parallel.

We use FFJORD (Grathwohl et al., 2019) as the baseline model for our CNFs. In addition, we use to two regularization terms introduced by RNODE (Finlay et al., 2020) to speed up the training of FFJORD models.

**Generation**: First, $\mathbf{z}_s, s = 1, \ldots, S$ are sampled from the latent noise distributions. Given an $S$-resolution model, CNF $g_s$ at resolution $s$ transforms the noise sample $\mathbf{z}_s$ to high-level information $\mathbf{y}_s$ conditioned on the immediate coarse image $\mathbf{x}_{s+1}$ (except $g_S$ which is unconditioned). $\mathbf{y}_s$ and $\mathbf{x}_{s+1}$ are then combined to form $\mathbf{x}_s$ as described in section 3.2 (see fig. 1). This process is repeated progressively from coarser to finer resolutions:

$$\mathbf{x}_S = g_S^{-1}(\mathbf{z}_S) \qquad s = S$$

$$\begin{cases} \mathbf{y}_s = g_s^{-1}(\mathbf{z}_s \mid \mathbf{x}_{s+1}) \\ \mathbf{x}_s = M(\mathbf{y}_s, \mathbf{x}_{s+1}) \end{cases} \quad s = S\text{-}1 \to 1 \quad (13)$$

## 4. Related work

Several prior works on normalizing flows — Glow (Kingma & Dhariwal, 2018), Hoogeboom et al. (2019a;b), Mint-Net (Song et al., 2019), MaCow (Ma et al., 2019), Durkan

et al. (2019); Chen et al. (2020), Flow++ (Ho et al., 2019), NanoFlow (Lee et al., 2020), Wavelet Flow (Yu et al., 2020), DenseFlow (Grcić et al., 2021) — build on RealNVP (Dinh et al., 2017). Although they achieve great results in terms of BPD and image quality, they nonetheless report results from significantly higher parameter (some with 100x!), and several times GPU hours of training.

Although our MRCNF model is similar to the recently published WaveletFlow (Yu et al., 2020), we generalize the notion of a multi-resolution image representation. Wavelet-Flow builds on the Glow (Kingma & Dhariwal, 2018) architecture, while ours builds on CNFs. WaveletFlow claims to have orthonormal transformation, our eq. (6) involves a unimodular transformation. Finally, WaveletFlow applies special sampling techniques to obtain better samples from its model. We have so far not used such techniques for generation, but we believe they can potentially help our models as well.

**"Multiple scales" in prior normalizing flows**: Normalizing flows (Dinh et al., 2017; Kingma & Dhariwal, 2018; Grathwohl et al., 2019) try to be "multi-scale" by transforming the input at one resolution in a smart way (squeezing operation) such that the width of the features progressively reduces. In contrast, our model stacks normalizing flows at multiple *resolutions* in an autoregressive fashion.

## 5. Experimental results

We train Multi-Resolution Continuous Normalizing Flow (MRCNF) and Multi-Resolution Continuous Normalizing Flow - Wavelet (MRCNF-Wavelet) models on the ImageNet (Deng et al., 2009) dataset at 32x32, 64x64, 128x128. We build on the code provided in (Finlay et al., 2020) (https://github.com/cfinlay/ffjord-rnode). In all cases, we train using *only one* NVIDIA RTX 2080 Ti GPU with 11GB.

At lower resolution spaces, we achieve comparable BPDs in lesser time with far fewer parameters than previous normalizing flows (and non flow-based approaches). However, the power of the multi-resolution formulation is more evident at higher resolutions: we achieve state-of-the-art BPD for ImageNet64 with significantly fewer parameters and lower time using only one GPU.

**Progressive training**: Since each resolution can be trained independently, we train an MRCNF model on ImageNet128 by training only the finest resolution (128×128) conditioned on the immediate coarser (64×64) images, and attach that to a 3-resolution 64×64 model. The resulting 4-resolution ImageNet128 model gives a BPD of **3.31** (Table 2) with just 2.74M parameters and 59 GPU hours of total training time.

*Table 1.* Bits-per-dimension (lower is better) of images for CIFAR10, ImageNet at 32×32, and ImageNet at 64×64, reported as the mean and standard deviation across the dataset. We also report the number of parameters in the models, and the time taken to train (in GPU hours). Most previous models use multiple GPUs for training, all our models were trained on only *one* GPU: NVIDIA RTX 2080 Ti 11GB. ‡As reported in (Ghosh et al., 2020). §Re-implemented by us. 'x': Fails to train. Blank spaces indicate unreported values. *RNODE (Finlay et al., 2020) used 4 GPUs to train on ImageNet64.

| | IMAGENET32 | | | IMAGENET64 | | |
|---|---|---|---|---|---|---|
| | BPD | PARAM | TIME | BPD | PARAM | TIME |
| **Flow-based Prior Work** | | | | | | |
| RealNVP | 4.28 | 46.0M | | 3.98 | 96.0M | |
| Glow | 4.09 | 66.1M | | 3.81 | 111.1M | |
| MintNet | 4.06 | 17.4M | | | | |
| MaCow | | | | 3.69 | 122.5M | |
| Flow++ | 3.86 | 169M | | 3.69 | 73.5M | |
| Wavelet Flow | 4.08 | 64M | | 3.78 | 96M | 822 |
| DenseFlow | 3.63 | | 310 | 3.35 | | 224 |
| **1-Resolution CNF** | | | | | | |
| FFJORD | 3.96‡ | 2.00M‡ | >5days‡ | x | | x |
| RNODE | 2.36‡ | 2.00M | 30.1‡ | 3.83* | 2.00M | 64.1* |
| | 3.49§ | 1.58M§ | 40.39§ | | | |
| FFJORD + STEER | 3.84 | 2.00M | >5days | | | |
| RNODE + STEER | 2.35 | 2.00M | 24.9 | | | |
| | 3.49§ | 1.58M§ | 30.07§ | | | |
| **(OURS) Multi-Resolution CNF (MRCNF)** | | | | | | |
| 2-resolution | 3.77 | 1.33M | 18.18 | - | - | - |
| 2-resolution | 3.78 | 6.68M | 17.98 | - | - | - |
| 3-resolution | 3.97 | 1.53M | 13.78 | 3.61 | 2.04M | 28.64 |

*Table 2.* Metrics for unconditional ImageNet128 generation.

| IMAGENET128 | BPD | PARAM | TIME |
|---|---|---|---|
| Parallel Multiscale (Reed et al., 2017) | 3.55 | | |
| SPN (Menick & Kalchbrenner, 2019) | 3.08 | 250M | |
| **(OURS) 4-resolution MRCNF** | 3.30 | 2.74M | 58.59 |

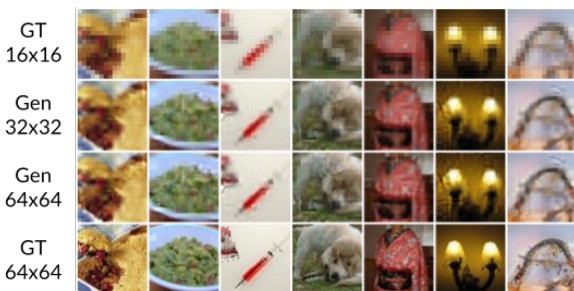

*Figure 2.* ImageNet: Example of super-resolving from ground truth 16×16 to 64×64. Top ground truth, middle generated, bottom ground truth.

# 6. Conclusion

We presented a Multi-Resolution approach to CNFs, which provides an efficient framework for likelihood calculations by training on a single GPU in lesser time with a significantly fewer parameters. We see a marked improvement in BPD for ImageNet64 and above.

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
