# OpenReview forum: "Improving Continuous Normalizing Flows using a Multi-Resolution Framework"
_ICML.cc/2021/Workshop/INNF — INNF+ 2021 poster_

### Official Review · Reviewer_VqwY · 2021-06-10

**Rating:** Accept
**Confidence:** 4

**Summary:**

The work proposes a multiresolution architecture, that reduces computational resources and time needed for training, allows for piecewise training while preserving the competitive level of quality.

Overall:
- The evaluation looks promising (training of closes competitor is 2.2 slower, with worse BPD) but lacks some details and ablations.
- Text is sometimes hard to read, but it looks fixable for camera-ready.

Detailed comments:


1. > Further, it is preferable that this transformation ..., and does not add too much change in terms of log-likelihood.

Why is that preferable? Is it a matter of computational complexity or smth else?

2. > We choose to perform a linear transformation taking into account the following properties: 1) volume preserving i.e. determinant is 1, 2) angle preserving, and 3) range preserving

    Is there intuition behind these choices? or is it more caused by a specific form of matrix M that is used later?

3. > Finding the rest of the parameters can be viewed as requiring four 3D vectors that are non-trivially equally spaced.

    Why do vectors need to be "equally spaced"? Can't any M of det 1, or even easily computable det be used? The cause of this remains unclear.

4. Section 3.2 absolutely needs its own illustrations.

5. I suggest to remove or make significantly shorter the section "2.2. Continuous Normalizing Flows" as it is not heavily used in the paper. "In principle, any generative model could be used to map between the multi-resolution image and noise." Space can be used to explain more details about the proposed model.

6. In experiments, it is interesting to see a comparison with RNODE while preserving training time e.g., reduce the number of gradient steps for RNODE, and increase LR (possibly).

7. It is interesting to see a visual comparison, as well as 128x128 samples.

**Justification For Rating:**

Results look interesting for community.

---

### Official Review · Reviewer_RAbY · 2021-06-12

**Rating:** Borderline Accept
**Confidence:** 3

**Summary:**

The paper proposes a new method for building continuous normalizing flows for image generation. The image is generated by using multi-resolution representation.The authors make a connection to classical computer-vision and signal processign methods based on wavelets.

**Justification For Rating:**

The paper is well-written. It is easy to follow. The illustrations help reading the paper. The experiments on Imagenet64 demonstrate that the proposed method allows for better BPD while using almost the same number of parameters and taking less time to train.

The experiments on Imagenet32 does not demonstrate the advantage of the proposed method over FFJORD for example. But as it was discussed in the paper, the power of the proposed method is better observed when the resolution is increased.

---

### Decision · Program_Chairs · 2021-06-15

**Decision:**

Accept (poster)

**Comment:**

This paper is on topic for the workshop and the reviews were overall positive. Therefore, we have accepted this paper. The paper claims state-of-the-art results on imagenet-64 in terms of bits per dimension, which seems inaccurate when checking against other methods on the website Papers With Code (https://paperswithcode.com/sota/image-generation-on-imagenet-64x64). Please adjust this claim accordingly, and take into account the reviewer's suggestions when preparing the camera ready version.